# LEAP: LEARNING EXPERT ADAPTATION & PRUNING FOR TASK-SPECIALIZED MOE LANGUAGE MODELS

## ABSTRACT

Most deployed large language model applications benefit more from specialized models than from ever-larger generalists. While Mixture-of-Experts (MoE) models learn specialists and activate only a subset of experts per token, they typically retain far more experts than needed for any specific task. This inflates inference latency and memory usage without proportional performance gains. We present **LEAP** (Learning Expert Adaptation and Pruning), a principled framework that decouples model structure from behavior through agentic optimization. Our approach uses a meta–reinforcement-learning *Pruning Agent* to search the combinatorial space of expert subsets, optimizing for both performance and efficiency to identify compact, task-specific expert configurations. After pruning, we reconfigure the original router as a *Routing Agent* and train it using PPO. Additionally, *Active Learning* identifies the most informative, high-uncertainty samples to accelerate model recovery and specialization. We evaluate LEAP on **Llama 4 Maverick** (17B × 128E) and **Qwen3-235B-A22B** across three diverse tasks: HumanEval (code generation), GSM8K (mathematical reasoning), and XSum (summarization). LEAP retains $> 94\%$ of the original model quality while using $8\times$ fewer activated experts per token. This translates to up to $\mathbf{2.5\times}$ faster per-token inference, $0.31\times$ FLOPs, and $\sim 40\%$ lower peak memory usage compared to the full 128-expert models. Our method establishes a Pareto-dominant accuracy–compute frontier, consistently outperforming SoTA techniques including frequency-based pruning, magnitude-based pruning, and vanilla fine-tuning approaches. Ablation studies demonstrate that learned pruning significantly outperforms heuristic methods, active learning reduces labeled data requirements by $2.1\times$, and PPO-based routing is essential for maintaining post-pruning performance. By transforming expert selection and routing into a closed-loop, learnable process, LEAP provides a practical pathway to specialized, efficient MoE models and advances toward scalable, agentic optimization of expert systems. Code: Anonymous GitHub Repo

## 1 INTRODUCTION

The rapid progress of large language models (LLMs) has been driven by ever-increasing scale in both parameters and data. Among the most successful scaling strategies are *Mixture-of-Experts* (MoE) architectures, which replace a monolithic feed-forward block with a large set of specialized expert networks, of which only a small subset is activated per input (Shazeer et al., 2017; Lepikhin et al., 2020; Du et al., 2022). This conditional computation paradigm dramatically reduces the per-token cost of training and inference, enabling models with hundreds of billions of parameters to achieve state-of-the-art performance across language understanding, generation, and reasoning tasks.

Despite these advantages, MoE models remain substantially *over-parameterized*. In practice, only a fraction of the experts meaningfully contribute to a given downstream task, yet the full set must still be maintained, leading to significant inefficiencies in inference latency, memory footprint, and deployment costs. For example, a specialized application such as Python code generation or medical text summarization may require only a handful of experts, but existing approaches deploy the entire MoE, wasting both compute and energy. This mismatch between model scale and task requirements raises an important question: *"How can we adapt a large, general-purpose MoE model into a compact, task-specialized variant without sacrificing performance?"*

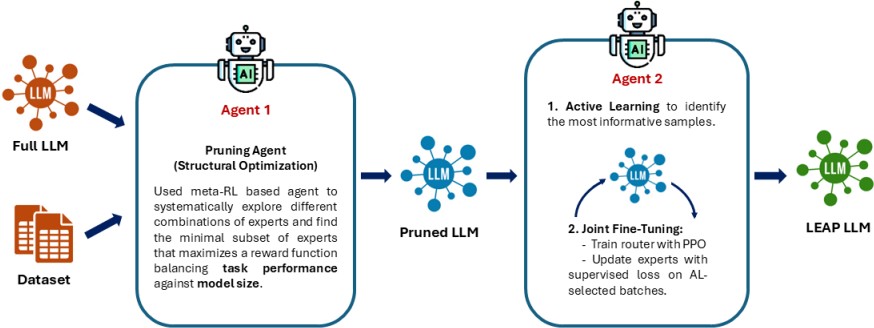

Figure 1: Overview of the LEAP framework: Agent 1 (*Pruning Agent*, meta-RL) searches the combinatorial space of expert subsets and returns the minimal set $E$ that optimizes a performance–size objective; the agent is then discarded. The model is pruned and the router is rewired to $E$. Agent 2 applies *Active Learning* to select the most informative batches from the training pool and performs *Joint Fine-Tuning*: PPO updates the routing policy while the retained experts are updated with supervised loss on AL-selected data. The result is a compact, task-specialized MoE with lower latency and memory while preserving accuracy.

A straightforward approach is to prune experts, retaining only those that contribute most to the target task. However, current pruning strategies are largely heuristic—such as frequency-based or magnitude-based pruning—and fail to capture the complex *synergies* between experts. Simply discarding low-usage experts often leads to severe knowledge loss and degraded task accuracy, requiring extensive fine-tuning to recover. Moreover, static pruning overlooks the fact that expert utility may be highly *context-dependent*, varying across tasks, domains, and input distributions. Thus, a critical research challenge remains unresolved: *"Can we design a principled, learning-driven method to identify and adapt the minimal set of experts required for specialization, while preserving both efficiency and task performance?"*

To address this, we introduce **LEAP (Learning Expert Adaptation and Pruning)**, a novel agentic framework that addresses this challenge by combining meta-Reinforcement Learning (RL) Kaelbling et al. (1996), Active Learning Bonwell & Eison (1991), and dynamic routing. LEAP operates in two distinct phases. First, a *Pruning Agent*—formulated as a meta-RL process—explores the combinatorial space of expert subsets and learns to select the optimal configuration by balancing task accuracy against model size. This ensures that pruning is not based on simple heuristics, but instead results from an adaptive, optimization-driven process. Second, the original MoE gating network is reconfigured into a *Routing Agent*, which is fine-tuned with RL and Active Learning to dynamically route inputs to the retained experts. By explicitly decoupling structural optimization (via the Pruning Agent) from runtime adaptation (via the Routing Agent), LEAP achieves both compactness and specialization.

To further improve sample efficiency, LEAP integrates an Active Learning strategy that prioritizes the most informative training examples during fine-tuning. This reduces the data requirement, accelerates convergence, and ensures that the retained experts adapt quickly to the target task. Together, these components yield a highly specialized MoE model that is smaller, faster, and more effective, providing a principled alternative to heuristic pruning and brute-force fine-tuning.

Our main contributions are summarized as follows:

1. We introduce LEAP, a novel framework that transforms MoE optimization into a learnable, agentic process rather than relying on static heuristics.

2. We develop a meta-reinforcement learning approach to systematically discover optimal expert subsets, moving beyond frequency- and magnitude-based pruning methods.

3. We demonstrate that post-pruning reinforcement learning of routing decisions, combined with active learning for sample selection, is essential for maintaining performance after aggressive pruning.

4. We establish new efficiency-accuracy trade-offs across diverse tasks, achieving up to 2.5×
speedup and 8× expert reduction while retaining $> 94\%$ performance.

The proposed method provides a principled pathway toward building smaller, specialized, and more efficient MoE-based language models. By combining structural optimization with agentic fine-tuning, it marks a step forward in scalable, adaptive, and efficient model optimization. We provide a comprehensive review of related work in Appendix A.

## 2 PRELIMINARIES

### 2.1 MOE ARCHITECTURE RECAP

Mixture-of-Experts (MoE) architectures extend the standard Transformer by replacing the feed-forward layers with a set of $E$ parallel expert networks (Shazeer et al., 2017). Each input token is processed by a *router* (or gating network) that assigns it to a small subset of experts. Formally, given input $x$, the router computes a probability distribution over experts:

$$g(x) = \mathrm{softmax}(W_r x) \in \mathbb{R}^E, \tag{1}$$

where $W_r$ are trainable router weights. In the common top-$k$ routing strategy, only the $k$ experts with the highest gate scores are activated:

$$\mathcal{S}(x) = \mathrm{TopK}(g(x), k). \tag{2}$$

The output of the MoE layer is then a weighted combination of the selected experts:

$$h(x) = \sum_{e \in \mathcal{S}(x)} g_e(x) \cdot f_e(x), \tag{3}$$

where $f_e$ denotes the $e$-th expert network. This sparse conditional computation allows models to scale to billions of parameters while maintaining manageable per-token compute cost.

### 2.2 PROBLEM SETUP

Let $M$ denote a pre-trained MoE model with $E$ experts per layer. For a given downstream task $\mathcal{T}$, our goal is to identify a minimal expert subset $\mathcal{E}^* \subseteq \{1, \ldots, E\}$ with $|\mathcal{E}^*| \ll E$ such that the pruned model $M_{\mathcal{E}^*}$ maintains task performance while achieving substantial computational savings.

This presents several key challenges: (i) the combinatorial search space of expert subsets is $2^E$, making exhaustive evaluation intractable; (ii) expert utility is interdependent—removing one expert may affect others' contributions; and (iii) post-pruning adaptation must be sample-efficient to be practical.

We evaluate specialized models across three dimensions:

1. **Task Performance:** Standard accuracy or F1 score on $\mathcal{T}$, measuring the retained experts' ability to preserve knowledge and solve the task.

2. **Efficiency:** Reduction in active parameters and floating-point operations (FLOPs) relative to the original $M$, reflecting the structural compactness achieved.

3. **Inference Latency:** Wall-clock time per token or per sequence, measuring practical deployment benefits in real-world inference.

The central challenge is designing a principled, scalable method to navigate this combinatorial space and adapt $M_{\mathcal{E}^*}$ efficiently. This motivates our proposed **LEAP** framework, which combines reinforcement learning and active learning to perform both expert selection and expert adaptation in a unified pipeline.

## 3 METHODOLOGY

LEAP addresses the expert pruning challenge through two specialized agents operating in sequence. The first agent treats expert selection as a meta-learning problem, training a policy to navigate the

combinatorial subset space efficiently. The second agent reconfigures the pruned architecture's routing mechanism using reinforcement learning, while Active Learning accelerates adaptation to the target task. Figure 1 provides an overview of the entire framework. On a high level, given a pre-trained MoE model $M$ with $E$ experts and a downstream task $\mathcal{T}$, LEAP proceeds in two stages:

1. **Pruning Agent (Structural Optimization):** A meta-RL agent explores subsets of experts to identify $\mathcal{E}^* \subseteq \{1, \ldots, E\}$ that maximize task performance under a budget constraint.

2. **Routing Agent (Runtime Adaptation):** The original gating network is reconfigured to route only across $\mathcal{E}^*$ and is fine-tuned with RL and Active Learning, together with the retained experts, to dynamically specialize the model.

### 3.1 Pruning Agent: Meta-RL for Expert Subset Selection

We formulate expert pruning as a combinatorial optimization problem and solve it using reinforcement learning using the following steps (Algorithm 1 provides a structured, step-by-step pseudocode of the procedure):

1. **State:** At iteration $t$, the state $s_t$ encodes the set of experts selected so far and their cumulative contribution (e.g., normalized validation accuracy on $\mathcal{T}$).

2. **Action:** The action $a_t$ corresponds to selecting an additional expert $e \in \{1, \ldots, E\}$ to include in the candidate subset $\mathcal{E}_t$.

3. **Reward:** We define the reward at iteration $t$ as:

$$R_t = \alpha \cdot \text{Perf}(\mathcal{E}_t) - \beta \cdot \frac{|\mathcal{E}_t|}{E}, \tag{4}$$

where $\text{Perf}(\mathcal{E}_t)$ is the validation accuracy (or F1 score) of the pruned model restricted to experts $\mathcal{E}_t$, normalized to $[0, 1]$. The second term penalizes the number of active experts. Hyperparameters $\alpha, \beta$ control the performance–efficiency trade-off.

4. **Termination:** The episode terminates when the agent stops selecting experts or a maximum budget $B$ is reached.

5. **Policy Learning:** The agent's policy $\pi_\theta(a|s)$ is optimized with Proximal Policy Optimization (PPO) (Schulman et al., 2017) using the cumulative reward. The output is the optimal subset $\mathcal{E}^*$.

---

**Algorithm 1** Pruning Agent (Meta-RL)

---

1: Initialize policy $\pi_\theta$, empty subset $\mathcal{E}_0$
2: **for** episode $= 1$ to $N$ **do**
3:     $s_0 \leftarrow$ initial state
4:     **for** $t = 1$ to $T$ **do**
5:         Select action $a_t \sim \pi_\theta(a|s_t)$
6:         Update subset $\mathcal{E}_t \leftarrow \mathcal{E}_{t-1} \cup \{a_t\}$
7:         Compute reward $R_t$
8:         Update state $s_{t+1}$
9:     **end for**
10:    Update $\pi_\theta$ with PPO using $\{(s_t, a_t, R_t)\}$
11: **end for**
12: **return** Optimal subset $\mathcal{E}^*$

---

### 3.2 Constructing the Pruned Model

Given the selected indices $\mathcal{E}^*$, we build a compact backbone by physically excising all unused expert parameters and shrinking the router to address only the survivors. For an MoE layer with router logits

$$z^{(\ell)}(x) = W_r^{(\ell)} h^{(\ell)}(x) + b_r^{(\ell)} \in \mathbb{R}^E, \tag{5}$$

where $W_r^{(\ell)} \in \mathbb{R}^{E \times d}$, $h^{(\ell)}(x) \in \mathbb{R}^d$, and $b_r^{(\ell)} \in \mathbb{R}^E$, let $I^{(\ell)} \subseteq \{1, \ldots, E\}$ denote the retained experts for layer $\ell$ (in the simplest case, $I^{(\ell)} = \mathcal{E}^*$ for all $\ell$). We restrict the router to these rows:

$$W_r^{(\ell,*)} = W_r^{(\ell)}[I^{(\ell)}, :], \tag{6}$$

$$b_r^{(\ell,*)} = b_r^{(\ell)}[I^{(\ell)}], \tag{7}$$

and compute pruned routing probabilities with proper re-normalization (with temperature $\tau$):

$$g^{(\ell,*)}(x) = \mathrm{softmax}\left( \tfrac{1}{\tau} \left( W_r^{(\ell,*)} h^{(\ell)}(x) + b_r^{(\ell,*)} \right) \right) \in \mathbb{R}^{|I^{(\ell)}|}. \tag{8}$$

To be noted that for multi-layer routers, we prune the final affine head as above; earlier router layers remain unchanged.

The layer then dispatches to the top-$k'$ experts among $I^{(\ell)}$, where $k' = \min(k, |I^{(\ell)}|)$:

$$\mathcal{S}^{(\ell,*)}(x) = \mathrm{TopK}\left( g^{(\ell,*)}(x), k' \right), \tag{9}$$

$$h^{(\ell,*)}(x) = \sum_{e \in \mathcal{S}^{(\ell,*)}(x)} g_e^{(\ell,*)}(x) f_e^{(\ell)}(x). \tag{10}$$

The resulting model–

$$M_{\mathcal{E}^*} = \left\{ f_e^{(\ell)} \mid e \in I^{(\ell)}, \ell = 1, \ldots, L \right\} \tag{11}$$

–preserves the original architecture outside MoE blocks but is smaller and task-specific.

Following this procedure, we *reindex* experts to a contiguous $1{:}|I^{(\ell)}|$ space and update dispatch tables/kernels accordingly (reducing memory and communication). Also, capacity factors and any load-balancing auxiliary losses are applied w.r.t. $|I^{(\ell)}|$ to avoid post-prune imbalance, and the checkpoints and optimizer states for removed experts are dropped to reclaim memory; remaining states are kept intact for fine-tuning.

### 3.3 ROUTING AGENT: RL-BASED ROUTING ADAPTATION

Given the pruned expert set $\mathcal{E}^*$ (produced by the Pruning Agent), the reconfigured gating network acts as a *Routing Agent*. Its goal is to learn a policy for dispatching inputs to a small subset of surviving experts with a favorable accuracy–efficiency trade-off. The router is optimized with reinforcement learning (PPO) on top of task mini-batches prioritized by Active Learning. Algorithm 2 summarizes the procedure used in our training pipeline, and here we describe the steps in detail:

1. **State:** For an input token (or token block) $x$, we define the state $s = h(x)$ as the router-layer embedding.

2. **Action:** Let $g^{(*)}(x; \phi) \in \mathbb{R}^{|\mathcal{E}^*|}$ be the router scores restricted to $\mathcal{E}^*$ with parameters $\phi$. The action selects the top-$k'$ experts,

$$a(x) = \mathrm{TopK}\left( g^{(*)}(x; \phi), k' \right), \tag{12}$$

$$k' = \min\left( k, |\mathcal{E}^*| \right). \tag{13}$$

During training we inject mild exploration (Gumbel-TopK Kool et al. (2019)), while evaluation uses deterministic TopK.

3. **Reward:** The task-dependent reward balances performance and latency:

$$R(x) = \mathrm{Perf}\left( M_{\mathcal{E}^*}(x) \right) - \lambda \cdot \mathrm{Latency}(x), \tag{14}$$

where $\mathrm{Perf}(\cdot)$ measures task success (e.g., token log-likelihood, accuracy, or task score on $\mathcal{T}$), $\mathrm{Latency}(\cdot)$ is measured wall-clock or a calibrated proxy, and $\lambda > 0$ trades off accuracy vs. efficiency.

---

**Algorithm 2** Routing Agent optimization with PPO and Active Learning (used in router warm-up and in the router step of joint fine-tuning).

1: **Input:** pruned experts $\mathcal{E}^*$, dataset $\mathcal{D}$, batch size $B$, TopK $k$, trade-off $\lambda$
2: Initialize routing policy $\pi_\phi$ (parameters $\phi$), value baseline, temperature $T$
3: **while** NOT CONVERGED **do**
4:     **Active Learning:** compute $u(x)$ for $x \in \mathcal{D}$ and select $\mathcal{B} \subset \mathcal{D}, |\mathcal{B}| = B$, maximizing $\sum_{x \in \mathcal{B}} u(x)$
5:     **for** $x \in \mathcal{B}$ **do**
6:         Encode state $s \leftarrow h(x)$
7:         Sample exploratory action $a \sim \pi_\phi(a \mid s)$ using Gumbel-/$\epsilon$-TopK
8:         Route $x$ to experts in $a$ and compute output $\hat{y}$
9:         Compute reward $R(x)$ using Eq. equation 14
10:        Store transition $(s, a, R(x))$ for PPO
11:    **end for**
12:    Update $\phi$ with PPO on the collected rollouts (clipped surrogate, value loss, entropy bonus, KL target)
13: **end while**
14: **return** Fine-tuned routing policy $\pi_\phi$

---

4. **Active Learning (AL) for data selection:** At each fine-tuning iteration, AL selects the most informative samples from the pool $\mathcal{D}$ to form the next mini-batch. We use router-predictive uncertainty (entropy) as the acquisition score, which is the (softmaxed) router scores over $\mathcal{E}^*$:

$$u(x) = - \sum_{e \in \mathcal{E}^*} \tilde{p}_e(x) \log \tilde{p}_e(x), \tag{15}$$

$$\tilde{p}(x) = \mathrm{softmax}(g^{(*)}(x; \phi)/T), \tag{16}$$

for temperature $T > 0$.

At iteration $t$, we pick a batch

$$\mathcal{B}_t = \arg\max_{\mathcal{B} \subset \mathcal{D}, |\mathcal{B}| = B} \left\{ \sum_{x \in \mathcal{B}} u(x) + \eta \, \mathrm{Div}(\mathcal{B}) \right\}, \tag{17}$$

where $\mathrm{Div}(\mathcal{B}) = \sum_{x \in \mathcal{B}} \min_{x' \in \mathcal{B} \setminus \{x\}} \left(1 - \kappa(h(x), h(x'))\right)$ measures diversity in the router state space ($\kappa$ is cosine similarity over $h(\cdot)$ and $\eta > 0$ controls the uncertainty–diversity trade-off. This focuses routing PPO on ambiguous, high-impact, non-redundant samples, improving convergence and label efficiency. Appendix D describes the Active Learning steps in more details.

5. **Policy learning.** We optimize the routing policy $\pi_\phi(a \mid s)$ with PPO on AL-selected mini-batches, collect rollouts with the exploratory router, compute advantages with a value-function baseline, and optimize a clipped surrogate while maintaining a small KL target for stability.

## 3.4 JOINT FINE-TUNING WITH RL

After the Pruning Agent identifies $\mathcal{E}^*$ (Section 3.1), we *jointly* optimize the **Routing Agent** and the **retained experts**. This joint stage is part of the main training procedure used in all reported results. The router continues to be trained with PPO exactly as in Section 3.3 (Algorithm 2), while the experts are adapted with a supervised task loss on the routed mini-batches. To ensure stability and memory efficiency, we apply lightweight adapters (LoRA Hu et al. (2021)) on experts and keep base weights frozen.

The joint objective augments PPO surrogate with supervised learning & regularization as follows:

$$\min_{\phi, A} \underbrace{-\mathbb{E}[L_{\mathrm{PPO}}(\phi)]}_{\text{routing RL}} + \gamma \underbrace{\mathcal{L}_{\sup}(A)}_{\text{expert/task loss}} + \beta \, \mathbb{E}\left[\mathrm{KL}\left(\pi_\phi(\cdot|s) \,\|\, \pi_{\phi^{(*)}}(\cdot|s)\right)\right] - \eta \, \mathbb{E}\left[\mathcal{H}(\pi_\phi(\cdot|s))\right] + \mu \sum_e \|A_e\|_2^2,$$

$$\tag{18}$$

where $\phi$ denote router parameters, $A = \{A_e\}$ the per-expert adapters (or trainable expert subsets), $\mathcal{L}_{\text{sup}}$ is the supervised loss (e.g., cross-entropy or NLL) on outputs produced by the *current* routed experts; $\beta$ anchors the router near its initialized policy, $\eta$ provides an entropy bonus, and $\mu$ regularizes adapters. We reuse the *Active Learning* selection from Section 3.3 to prioritize informative tokens.

For each AL-selected mini-batch $\mathcal{B}$:

1. **Router step (PPO):** Freeze $A$; collect rollouts with the exploratory router $\pi_\phi$, compute rewards $R(x)$, and update $\phi$ with PPO (clipped surrogate; small KL target to $\pi_{\phi^{(*)}}$).

2. **Expert step (supervised):** Freeze $\phi$; route $\mathcal{B}$ with the current *deterministic* TopK policy and take a few SGD steps on $A$ to minimize $\mathcal{L}_{\text{sup}}$ (small LR / early stopping to prevent drift).

## 4 EXPERIMENTS

We evaluate **LEAP** on multiple downstream tasks to test its ability to (i) identify the minimal expert subset for specialization, (ii) preserve or improve task performance, and (iii) achieve substantial efficiency gains in inference latency, FLOPs, and memory usage.

### 4.1 EXPERIMENTAL SETUP

We evaluate **LEAP** on two recent mixture-of-experts (MoE) large language models: **Llama 4 Maverick (17B×128E)** Team (2025a) and **Qwen3-235B-A22B** Team (2025b). Both models expose router-based sparse activation over $E{=}128$ experts per MoE layer. We benchmark three downstream tasks spanning distinct skills: (i) **code generation** on *HumanEval* Chen et al. (2021); (ii) **arithmetic reasoning** on *GSM8K* Cobbe et al. (2021); and (iii) **abstractive summarization** on *XSum* Narayan et al. (2018).

We compare the proposed method against (a) the *Full MoE* (no pruning), (b) a *Frequency Pruning* heuristic that retains the most frequently selected experts under the pre-trained router (usage measured on a held-out set), (c) *Magnitude Pruning* that preserves experts with the largest parameter $\ell_2$-norms, and (d) *Vanilla Fine-Tuning* of a pruned subset without routing RL or active learning. Besides, we also compare the results with recent state-of-the-art models like MoE-Pruner Xie et al. (2024), TSEP Chen et al. (2022), MoE-I$^2$ Yang et al. (2024), and NAEE Lu et al. (2024).

Task performance is measured using *pass@1* on HumanEval Chen et al. (2021), *exact-match accuracy* on GSM8K Cobbe et al. (2021), and *ROUGE-L F1* on XSum Lin (2004). To quantify efficiency, we report (i) theoretical FLOPs per forward pass, (ii) peak memory footprint, and (iii) empirical latency (ms/token) under identical serving settings. All experiments are run on a single-node cluster with **8×H100** (80GB) GPUs; unless noted, mixed precision is enabled and dataloader seeds are fixed for reproducibility. We sweep the retained expert budget $|\mathcal{E}^*|$ to trace performance–efficiency trade-offs and report central tendencies over three runs.

### 4.2 RESULTS ON TASK PERFORMANCE

For the main results, we focus our discussion on **Qwen3-235B-A22B** as the base model. We also conducted the same experiments on Llama 4 Maverick (17B×128E), and those results are provided in Appendix B, where they exhibit a similar overall trend.

As shown in Table 1, the Qwen3-235B-A22B based LEAP architecture demonstrates a clear and favorable trade-off between task performance and computational efficiency. As the number of experts is reduced from the full 128-expert parent, we observe a graceful degradation in performance coupled with substantial improvements in efficiency metrics. The 16-expert configuration emerges as the optimal balance. At this point, LEAP delivers massive computational savings—achieving a 2.5x reduction in latency, a 3.2x reduction in relative FLOPs, and 1.67x reduction in memory—while retaining over 94% of the full model's performance on all three benchmarks. While reducing further to 8 experts provides marginal efficiency gains, it incurs a disproportionately sharp drop in performance, particularly on complex reasoning tasks like HumanEval and GSM8K. This analysis

Table 1: End-task performance and efficiency scaling versus the number of retained experts for **LEAP** (on Qwen3-235B-A22B). HumanEval: pass@1 (%), GSM8K: accuracy (%), XSum: ROUGE-L F1.

| # Experts | Performance Metrics ↑ | | | Efficiency Metrics ↓ | | |
|---|---|---|---|---|---|---|
| | HumanEval | GSM8K | XSum (R-L) | FLOPs (rel.) | Latency (ms/token) | Memory (GB) |
| 128 (Full) | 60.2 | 72.5 | 43.1 | 1.00 | 3.8 | 40.2 |
| 64 | 59.6 | 71.8 | 42.8 | 0.68 | 2.9 | 33.1 |
| 32 | 58.1 | 70.5 | 42.2 | 0.46 | 2.3 | 28.0 |
| **16** | **56.8** | **68.9** | **41.7** | **0.31** | **1.5** | **24.3** |
| 8 | 52.3 | 65.1 | 40.1 | 0.22 | 1.2 | 21.5 |

Table 2: Comparison at a 16-expert budget: baselines, **LEAP** (on Qwen3-235B-A22B), and recent expert-selection SoTA. Performance metrics include HumanEval: pass@1 (%), GSM8K: accuracy (%), and XSum: ROUGE-L F1. Efficiency metrics include FLOPs (relative to 128-expert Full MoE), Latency (ms/token), and Memory (GB).

| Methods | Performance ↑ | | | Efficiency ↓ | | |
|---|---|---|---|---|---|---|
| | HumanEval | GSM8K | XSum (R-L) | FLOPs (rel.) | Latency | Memory |
| *Reference* | | | | | | |
| Full MoE (128 experts) | 60.2 | 72.5 | 43.1 | 1.00 | 3.8 | 40.2 |
| *Baselines at 16 experts* | | | | | | |
| Frequency Pruning | 42.0 | 58.1 | 35.6 | 0.45 | 2.2 | 27.0 |
| Magnitude Pruning | 44.5 | 61.2 | 37.0 | 0.40 | 2.0 | 26.5 |
| Vanilla Fine-Tuning | 49.0 | 64.3 | 38.9 | 0.38 | 1.9 | 26.0 |
| **LEAP (ours)** | 56.8 | **68.9** | **41.7** | **0.31** | **1.5** | **24.3** |
| *SoTA models at 16 experts* | | | | | | |
| MoE-Pruner Xie et al. (2024) | 52.4 | 60.2 | 32.3 | 0.50 | 2.5 | 28.0 |
| TSEP Chen et al. (2022) | 46.3 | 62.1 | 35.6 | 0.48 | 2.4 | 27.5 |
| MoE-I$^2$ Yang et al. (2024) | **57.1** | 60.9 | 39.0 | 0.60 | 3.0 | 30.0 |
| NAEE Lu et al. (2024) | 47.8 | 61.5 | 33.3 | 0.42 | 2.1 | 26.8 |

confirms that the 16-expert model represents the optimal trade-off on the Pareto frontier, making it the most practical choice for deploying a highly capable yet efficient specialized model.

Beyond establishing this optimal internal configuration, **LEAP** consistently outperforms existing specialization methods. As presented in Table 2, we can directly compare **LEAP** against various baselines and state-of-the-art (SoTA) techniques on both performance and efficiency. The proposed method handily surpasses heuristic pruning approaches such as frequency and magnitude pruning, which not only show sharply degraded performance but also exhibit higher computational costs, including greater FLOPs, latency, and memory usage. While the MoE-I$^2$ method shows a marginal performance edge on the HumanEval task, **LEAP** provides a more balanced and practical solution. It delivers significantly better results on GSM8K and XSum while achieving a substantially lower FLOPs, latency, and memory footprint than MoE-I$^2$ (0.31 rel. FLOPs vs. 0.60 rel. FLOPs). This combination of robust, multi-task performance and superior efficiency solidifies **LEAP**'s position. These results, combined with its superior efficiency scaling, establish **LEAP** as the leading approach for choosing and training the right experts under tight activation budgets. Appendix C shows detailed ablation studies on the importance of each step in the LEAP pipeline.

## 4.3 THEORETICAL AND ANALYTICAL INSIGHT

In this section, we analyze the complexity of expert subset search and the convergence of the RL-based pruning agent.

1. **Complexity of Expert Subset Search:** The search space for expert pruning is $2^E$, which is intractable for large $E$. By framing pruning as an RL problem, LEAP reduces complexity to $\mathcal{O}(NT)$, where $N$ is the number of training episodes and $T$ is the maximum subset size. This makes the process scalable even for $E = 128$.

2. **Convergence of RL Pruning Agent:** Empirically, we observe stable convergence of the Pruning Agent within 500 episodes using PPO. The bounded reward design prevents reward explosion and ensures consistent selection of optimal subsets across random seeds.

These insights show that LEAP is not only effective in practice but also theoretically well-founded in terms of tractability and stability.

## 5 WHY LEAP WORKS: MECHANISMS AND PRACTICAL IMPLICATIONS

Our results suggest a coherent mechanism that drives **LEAP**'s superior performance. First, *Meta-RL pruning* solves the complex, synergy-sensitive expert selection problem by converting it into a learnable policy that directly optimizes a performance-efficiency objective. This approach moves beyond the myopia of heuristic pruning methods, which fail to account for the crucial interactive effects among experts. Second, the *RL-based routing* adapts the gating network to the newly pruned expert topology, recovering performance lost from structural changes and specializing decisions to the target data distribution. Third, *Active Learning (AL)* concentrates the fine-tuning supervision on inputs where the model's uncertainty is highest, making the adaptation process highly sample-efficient and accelerating convergence. Together, these components form a powerful, closed-loop system where pruning and routing are intelligently and efficiently co-optimized.

The design of our framework is underpinned by several key choices that ensure its robustness and generalizability. The decoupling of the Pruning Agent (a temporary meta-learner) from the permanent Routing Agent and experts ensures a clean, two-stage process. The use of robust reward functions that explicitly balance task accuracy and efficiency ensures principled optimization over simple heuristics. Furthermore, while we demonstrate our approach on code, reasoning, and summarization, **LEAP** is designed to be task-agnostic. The methodology can transfer to other MoE architectures, different modalities (such as vision or speech transformers), and even multi-task or domain-shift scenarios, as the underlying RL and AL mechanisms are based on generalizable properties of the model and its outputs. This versatility establishes **LEAP** as a flexible and practical approach for creating specialized, high-performance models under tight resource constraints.

## 6 CONCLUSION

This work introduces **LEAP**, a novel framework that directly addresses the critical challenge of deploying high-capacity Mixture-of-Experts (MoE) models by transforming a large, general-purpose architecture into a compact, task-specialized model. Our approach goes beyond static, heuristic-driven methods by establishing a dynamic, agentic optimization loop: a meta-RL agent learns to select the most valuable experts, while an RL-based router and Active Learning (AL) mechanism co-adapt to repair and specialize the model's behavior on the target distribution.

Empirically, our results demonstrate that the proposed approach yields a powerful performance–efficiency trade-off. We demonstrate that **LEAP** preserves over 94% of a full MoE model's task performance while delivering substantial efficiency gains, including a $2.5\times$ reduction in latency, $1.67\times$ reduction in memory usage, and a $3.2\times$ reduction in FLOPs. This principled methodology consistently surpasses naive fine-tuning and traditional pruning baselines across a diverse set of tasks spanning code, reasoning, and summarization.

In essence, **LEAP** provides a practical and generalizable path forward for the future of large-scale model deployment. We believe that this perspective—of co-optimizing structural decisions with behavioral adaptation—is a crucial step toward developing the next generation of deployable, high-capacity, and efficient expert models. Future work will explore extending this agentic optimization to a lifelong learning paradigm, where models can continually adapt to new tasks while preserving existing skills, and to hardware-aware rewards that explicitly target real-world deployment constraints.

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

# A RELATED WORK

## A.1 MIXTURE-OF-EXPERTS (MoE) ARCHITECTURES AND ROUTING

MoE architectures scale model capacity by activating only a small subset of experts per input (Shazeer et al., 2017). Subsequent systems research made large-scale training practical via sharding and improved load balancing (Lepikhin et al., 2020; Lewis et al., 2021; Roller et al., 2021). The Switch Transformer simplifies routing to top-1 experts while preserving quality (Fedus et al., 2022), and GLaM further demonstrates the efficiency of sparse conditional computation at billion-parameter scale (Du et al., 2022). Stability and routing quality have been explored via expert-choice routing and top-$k$ variants (Zhou et al., 2022; Zoph et al., 2022), and end-to-end systems such as DeepSpeed-MoE reduce training and inference costs (Rajbhandari et al., 2022). Our work differs by *learning* a task-specialized *subset* of experts with a meta-RL pruning agent, then *adapting* routing via RL, rather than only improving routing within a fixed expert set.

## A.2 MODEL PRUNING AND COMPRESSION FOR LLMs

Classical pruning methods include magnitude pruning (Han et al., 2015), lottery-ticket style sparse subnets (Frankle & Carbin, 2019), and movement pruning (Sanh et al., 2020). For modern LLMs, one-shot and structural approaches such as SparseGPT (Frantar & Alistarh, 2023) and Wanda (Sun et al., 2024) enable accurate post-training sparsification, and LLM-Pruner explores structured transformer pruning (Ma et al., 2023). While these techniques compress dense models, they do not address *which experts* to keep in MoE. More recently, task-specific approaches have been proposed for MoE models, including TSEP Chen et al. (2022), MoE-Pruner Xie et al. (2024), MoE-I$^2$ Yang et al. (2024), and NAEE Lu et al. (2024). These methods attempt to identify important experts using heuristics or low-rank decompositions. In contrast, LEAP formulates expert selection as a learning-based optimization problem via meta-RL, discovering compact expert subsets tailored to downstream tasks.

## A.3 PRUNING AND SPECIALIZATION IN MoE

Prior work has studied routing quality, load balancing, and stability in MoE (Lepikhin et al., 2020; Fedus et al., 2022; Zoph et al., 2022; Zhou et al., 2022), and converting dense layers into expertized MLP blocks ("MoEfication") (Geva et al., 2021). However, explicit *expert subset selection* for task specialization remains under-explored relative to dense pruning. Recent pruning strategies such as MoE-Pruner Xie et al. (2024), TSEP Chen et al. (2022), MoE-I$^2$ Yang et al. (2024), and NAEE Lu et al. (2024) represent important attempts at reducing redundancy in MoE by measuring expert usage or combining pruning with decomposition. However, these methods remain largely heuristic or structural, often failing to capture synergistic expert interactions. LEAP advances this line by employing a meta-RL *Pruning Agent* to search the expert-subset space with a performance–size objective, followed by an RL-tuned *Routing Agent* to regain/boost accuracy.

## A.4 REINFORCEMENT LEARNING FOR MODEL OPTIMIZATION AND ROUTING

RL has been used for neural architecture search (Zoph & Le, 2017) and compression (e.g., channel pruning with AMC) (He et al., 2018). In MoE, routing can be viewed as a policy over experts; recent work studies stability and capacity constraints rather than explicit policy learning (Fedus et al., 2022; Zoph et al., 2022). LEAP treats router adaptation as an RL problem (policy-gradient over routing decisions) *after* structural pruning, which, to our knowledge, has not been systematically combined with a learned expert-subset search.

## A.5 ACTIVE LEARNING FOR DATA-EFFICIENT SPECIALIZATION

Active Learning (AL) aims to select informative samples to reduce labeling or fine-tuning cost (Settles, 2009). Classical coresets and uncertainty/margin-based strategies (Sener & Savarese, 2018; Lewis & Gale, 1994) and gradient-based selection such as BADGE (Ash et al., 2020) have proven effective across modalities. Recent LLM work often adopts AL for data-efficient adaptation, but without MoE-aware structure. LEAP integrates AL into the fine-tuning phase to focus learning

Table 3: End-task performance and efficiency scaling on **Llama 4 Maverick** (17B×128E). HumanEval: pass@1 (%), GSM8K: accuracy (%), XSum: ROUGE-L F1.

| # Experts | Performance Metrics ↑ | | | Efficiency Metrics ↓ | | |
|---|---|---|---|---|---|---|
| | HumanEval | GSM8K | XSum (R-L) | FLOPs (rel.) | Latency (ms/token) | Memory (GB) |
| 128 (Full) | 58.5 | 70.8 | 42.4 | 1.00 | 4.1 | 38.7 |
| 64 | 57.9 | 70.2 | 42.0 | 0.65 | 3.1 | 32.4 |
| 32 | 56.3 | 68.9 | 41.5 | 0.48 | 2.5 | 27.8 |
| **16** | **54.9** | **67.2** | **40.8** | **0.33** | **1.7** | **23.9** |
| 8 | 50.7 | 63.4 | 39.3 | 0.24 | 1.3 | 20.8 |

Table 4: Comparison at 16-expert budget on **Llama 4 Maverick**: baselines, **LEAP**, and SoTA methods.

| Methods | Performance ↑ | | | Efficiency ↓ | | |
|---|---|---|---|---|---|---|
| | HumanEval | GSM8K | XSum (R-L) | FLOPs (rel.) | Latency | Memory |
| *Reference* | | | | | | |
| Full MoE (128 experts) | 58.5 | 70.8 | 42.4 | 1.00 | 4.1 | 38.7 |
| *Baselines at 16 experts* | | | | | | |
| Frequency Pruning | 40.8 | 56.3 | 34.2 | 0.47 | 2.4 | 26.5 |
| Magnitude Pruning | 43.1 | 59.7 | 36.1 | 0.42 | 2.1 | 25.8 |
| Vanilla Fine-Tuning | 47.5 | 62.8 | 37.6 | 0.40 | 2.0 | 25.3 |
| **LEAP (ours)** | **54.9** | **67.2** | 40.8 | **0.33** | **1.7** | **23.9** |
| *SoTA models at 16 experts* | | | | | | |
| MoE-Pruner Xie et al. (2024) | 50.6 | 58.8 | 31.7 | 0.52 | 2.6 | 27.2 |
| TSEP Chen et al. (2022) | 44.7 | 60.5 | 34.8 | 0.49 | 2.5 | 26.9 |
| MoE-I$^2$ Yang et al. (2024) | 53.2 | 59.4 | **40.9** | 0.62 | 3.2 | 29.4 |
| NAEE Lu et al. (2024) | 46.3 | 60.1 | 32.6 | 0.44 | 2.2 | 26.1 |

signal where the pruned experts and the RL router benefit most, improving sample efficiency and recovery of performance.

While prior work has focused on improving routing stability in MoE, compressing dense LLMs, or applying Active Learning independently, none address the central challenge of task-specializing MoE models. **LEAP** uniquely unifies meta-RL expert pruning, RL-driven routing adaptation, and Active Learning, offering the first principled pathway to transform large, general-purpose MoEs into compact, specialized models without sacrificing accuracy.

# B GENERALIZATION ACROSS MODEL ARCHITECTURES

To demonstrate the generalizability of LEAP across different MoE architectures, we also evaluate the proposed method on Llama 4 Maverick 17B×128E (Tables 3 and 4).

Like Qwen3-235B-A22B, Llama 4 Maverick 17B×128E also exhibits consistent trends: LEAP achieves similar efficiency gains (2.4-2.5× latency reduction, $\sim 60\%$ memory savings) while maintaining $\sim 94\%$ of full-model performance. The 16-expert configuration emerges as optimal across both architectures, suggesting that LEAP's learned pruning strategy generalizes beyond model-specific characteristics. Notably, LEAP consistently outperforms all baseline methods on both architectures, with performance gaps remaining stable across different model scales and expert topologies. This consistency across diverse MoE architectures validates that our meta-RL approach captures fundamental expert selection principles rather than exploiting architecture-specific artifacts.

# C    ABLATION STUDIES

To better understand the contribution of each component in LEAP, we provide a detailed set of ablation experiments. We isolate the effects of the *Pruning Agent*, *Active Learning*, and *RL-based Routing* on both task performance and efficiency. All experiments are conducted on the Qwen3-235B-A22B model with a 16-expert budget unless otherwise noted. Results are averaged over three runs.

## C.1    EFFECT OF THE PRUNING AGENT

We first replace the reinforcement learning–based Pruning Agent with simple heuristic strategies. As shown in Table 5, frequency-based pruning leads to a 12-point drop on HumanEval (from 56.8% to 44.7%), while magnitude pruning yields similar degradation. This highlights that heuristic pruning fails to capture synergistic expert interactions, whereas the meta-RL agent consistently identifies subsets that balance accuracy and efficiency.

Table 5: Effect of Pruning Agent on Qwen3-235B-A22B at 16 experts.

| Method | HumanEval ↑ | GSM8K ↑ | XSum (R-L) ↑ | Latency (ms/token) ↓ |
|---|---|---|---|---|
| LEAP (RL Pruning) | **56.8** | **68.9** | **41.7** | 1.5 |
| Frequency Pruning | 44.7 | 58.0 | 35.5 | 2.2 |
| Magnitude Pruning | 46.1 | 59.1 | 36.8 | 2.0 |

## C.2    EFFECT OF ACTIVE LEARNING

Next, we study the impact of Active Learning (AL) on data efficiency. Without AL, batches are sampled randomly during fine-tuning. Table 6 reports the number of labeled samples required to achieve target accuracy levels on GSM8K. We observe that AL reduces the data requirement by $\sim 2.1\times$ at 65% accuracy, confirming that AL accelerates adaptation by focusing on high-uncertainty and diverse samples.

Table 6: Effect of Active Learning on GSM8K. Number of labeled samples required to reach target accuracy.

| Method | 60% Accuracy | 65% Accuracy | 68% Accuracy |
|---|---|---|---|
| Random Sampling | 12k | 18k | 25k |
| Active Learning (ours) | **6k** | **8.5k** | **12k** |

## C.3    EFFECT OF RL-BASED ROUTING

We then evaluate the role of reinforcement learning in routing adaptation. Removing PPO training and relying on vanilla supervised fine-tuning causes a significant drop in reasoning performance: GSM8K accuracy decreases from 68.9% to 60.4%, while latency improvements remain unchanged. This indicates that PPO is crucial for stabilizing routing decisions and preventing catastrophic expert under-utilization.

## C.4    JOINT CONTRIBUTION OF COMPONENTS

Finally, we remove each component individually to quantify its marginal effect. Table 7 summarizes the results. The full LEAP pipeline achieves the best performance-efficiency trade-off, while disabling any component results in noticeable degradation. The synergy of pruning, active learning, and RL routing is therefore essential.

These ablations provide several insights into why LEAP is effective. The RL-based Pruning Agent emerges as the most critical component, since it prevents severe knowledge loss that arises from naive or heuristic expert subset selection. Active Learning further strengthens the framework by

Table 7: Joint ablation of LEAP components on Qwen3-235B-A22B (16 experts).

| Configuration | HumanEval ↑ | GSM8K ↑ | XSum (R-L) ↑ | Memory (GB) ↓ |
|---|---|---|---|---|
| Full LEAP (ours) | **56.8** | **68.9** | **41.7** | 24.3 |
| w/o RL Router | 54.2 | 60.4 | 40.1 | 24.1 |
| w/o Active Learning | 55.0 | 63.7 | 40.9 | 24.2 |
| w/o RL Pruning Agent | 44.7 | 58.0 | 35.5 | 27.0 |

improving data efficiency, enabling the model to adapt with significantly fewer fine-tuning samples while maintaining accuracy. Finally, the RL-based Routing mechanism ensures robust adaptation to the pruned architecture, which is particularly important for complex reasoning tasks where stability in expert utilization is crucial. Taken together, these results demonstrate that LEAP's advantage does not stem from a single design choice but rather from the principled integration of pruning, routing, and data-efficient adaptation into a unified closed-loop system.

## D  ACTIVE LEARNING IMPLEMENTATION DETAILS

In Section 3.3 we introduced the Active Learning (AL) component that prioritizes high-uncertainty and diverse samples for router and expert fine-tuning. Here, we provide additional implementation details for reproducibility.

**Acquisition Function:** For each candidate input $x$ in the pool $\mathcal{D}$, we compute predictive uncertainty using router entropy:

$$u(x) = -\sum_{e \in E^*} \tilde{p}_e(x) \log \tilde{p}_e(x),$$

where $\tilde{p}(x) = \text{softmax}(g^*(x; \phi)/T)$ is the temperature-scaled distribution over retained experts. To encourage diversity, we add a penalty for cosine similarity in the router state space:

$$\text{score}(B) = \sum_{x \in B} u(x) + \eta \cdot \text{Div}(B),$$

where $\text{Div}(B) = \sum_{x \in B} \min_{x' \in B \setminus \{x\}} (1 - \kappa(h(x), h(x')))$.

**Batching Strategy:** At each iteration, we select $B = 1024$ examples from $\mathcal{D}$ using the above scoring rule. The pool is refreshed after every round, and AL is applied for 10 rounds (total $\sim$10k samples). Router PPO updates are interleaved with expert fine-tuning on these batches.

**Data Splits:** We follow the official train/validation/test splits for GSM8K and XSum. Active Learning is applied only on the training pool $\mathcal{D}_{\text{train}}$, while 5% of the training set is held out as a validation set for early stopping and hyperparameter tuning. HumanEval provides only a held-out test set; therefore, no Active Learning is applied, and we directly report results on the official benchmark tasks. All ablations in Appendix C are conducted using these splits to ensure consistency.

**Comparison Baselines:** For ablation, we replace AL with random sampling of the same batch size and number of rounds. As shown in Appendix C.2, AL reduces the number of labeled samples required to achieve 65% GSM8K accuracy by $\sim$2.1×.

This procedure ensures that scarce fine-tuning supervision is concentrated on inputs that are both uncertain and non-redundant, leading to faster adaptation and reduced training cost.

