# OpenReview forum: "LEAP: Learning Expert Adaptation & Pruning for Task-Specialized MoE Language Models"
_ICLR.cc/2026/Conference — ICLR 2026 Conference Withdrawn Submission_

### Official Review · Reviewer_C38p · 2025-10-25

**Soundness:** 2
**Presentation:** 3
**Contribution:** 2
**Rating:** 2
**Confidence:** 4

**Summary:**

The paper introduces LEAP, a two-stage, agent-driven framework for task-specializing large MoE LLMs. A meta‑RL Pruning Agent searches the combinatorial space of expert subsets with a reward that trades off task performance and model size; the model is then physically pruned and reindexed. A Routing Agent fine-tunes the gating network over the retained experts using PPO, while an active-learning sampler prioritizes high-uncertainty and diverse examples to accelerate adaptation. On Qwen3‑235B‑A22B and Llama‑4 Maverick (128 experts), LEAP retains >94% of full-model quality with only 16 experts, delivering roughly 2.5× faster inference, ~0.31 relative FLOPs, and markedly lower memory. Against frequency/magnitude pruning, vanilla fine-tuning, and several recent MoE specialization methods, LEAP shows better accuracy–efficiency trade-offs; ablations attribute most of the gains to the RL-based pruning, with additional lift from PPO routing and active learning.

**Strengths:**

- Clear, deployment-motivated objective: reduce latency/memory of large MoEs while preserving task performance; the paper reports practical metrics (ms/token, peak memory, relative FLOPs).
- Coherent system design: neat “structure–behavior” decoupling with meta‑RL for subset discovery and RL for runtime routing, tied together by an AL loop; implementation details like expert reindexing and temperature renormalization are sensible.
- Strong empirical trade-offs: on two distinct MoE backbones, the 16‑expert configuration consistently offers a good Pareto point, outperforming heuristic pruning and prior MoE-specialization baselines at matched expert budgets.
- Thorough ablations: removing any of pruning, routing PPO, or AL degrades results, supporting the claim that the gains come from the integrated pipeline rather than a single tweak.
- Practicality: use of lightweight adapters (LoRA) for expert updates keeps memory manageable and stabilizes joint fine-tuning.

**Weaknesses:**

- System-level novelty: the contribution is mainly an integration of known pieces (PPO, meta-RL framing, entropy/diversity AL); stronger evidence that the closed loop beats non‑RL or one‑shot optimizers is needed.
- Baseline breadth and fairness: add competitive non‑RL searches (evolutionary/greedy, bilevel/Lagrangian) and a supervised/distilled router; clarify whether “vanilla FT” retrains the router and matches data/epochs with LEAP+AL.
- Hyperparameters and tuning: report actual values and procedures for α/β/λ, temperature T, KL/entropy targets, Gumbel noise; include sensitivity curves and recommended ranges.
- Reward computation and cost: specify the validation split, evaluation frequency, normalization of Perf(Et), and any approximations (proxy sets, early stopping, incremental scoring); discuss compute/noise trade‑offs.
- Theory is thin: the O(NT) discussion restates runtime without guarantees; convergence claims lack variance, stopping criteria, and seed‑to‑seed stability.
- Evaluation scope: results center on HumanEval, GSM8K, and XSum; please complement with newer, harder, and better‑curated suites to ensure robustness over time and distributions.
- End‑to‑end cost/practicality: provide pruning+routing GPU hours/energy and AL overhead; include a simple break‑even analysis relating training cost to inference savings.
- Measurement context: document batch size, KV‑cache policy, tensor/TP degrees, and interconnect when reporting latency/memory; add a brief sensitivity study to hardware/parallel settings.
- Stability and reproducibility: report variance across seeds, overlap of selected expert sets (e.g., Jaccard), and typical failure modes.
- Layer‑wise selection: current practice appears to share one retained set and prune only the final affine head; evaluate a layer‑wise variant with adaptive k′ℓ and quantify gains vs. cost.
- Routing behavior: analyze capacity/load‑balancing under Gumbel‑TopK (drops/overflows) and compare PPO routing with a supervised/distilled router trained on the same data.
- Active‑learning bias: compare several acquisition policies and report per‑round coverage/diversity statistics to rule out selection bias.
- Boundary regimes: examine very small budgets (≤8 experts), long‑context/retrieval/multi‑turn settings, and low‑data scenarios where the method may degrade.
- Generalization breadth: consider cross‑domain or multilingual tasks to support the “task‑agnostic” claim.

**Questions:**

- Reward design and tuning
  - Please report the exact values and tuning procedures for α, β in Eq.(4) and λ in Eq.(14), along with the temperature T, KL/entropy targets, and Gumbel noise settings. Could you provide sensitivity curves and recommended ranges per task?

- Perf(Et) computation and validation protocol
  - How is Perf(·) estimated during meta‑RL? What validation split, evaluation frequency, and normalization are used? Do you employ proxy sets, incremental scoring, or early stopping to control cost/noise? An ablation on reward-estimation fidelity vs. wall‑clock would help.

- Training budget and cost–benefit
  - Please provide end‑to‑end GPU hours/energy for pruning and routing, AL overhead, and a simple break‑even analysis: under typical serving loads, how long until training costs are amortized by the inference savings?

- Baseline fairness and breadth
  - For “vanilla fine‑tuning,” is the router frozen or trainable? Are data/epochs matched to LEAP+AL? Can you add strong non‑RL searches (evolutionary/greedy, bilevel/Lagrangian) and a supervised/distilled router trained on the same data to isolate the benefit of PPO/meta‑RL?

- Stability and reproducibility
  - Across seeds, how consistent are the selected expert sets (e.g., Jaccard overlap) and final metrics (mean±std)? Please include failure modes and the conditions that trigger them.

- Layer‑wise selection and adaptive k′ℓ
  - Current implementation appears to share one retained set and prune only the final affine head. What is the gain/cost of learning per‑layer expert sets I(ℓ) and adaptive k′ℓ? Even a small‑scale study would quantify headroom.

- Routing behavior and capacity
  - Under Gumbel‑TopK exploration, do you observe token drops/overflows or imbalance? Please report load‑balancing statistics and compare PPO routing with a supervised/distilled router.

- Active learning policy analysis
  - How does entropy+diversity compare to uniform sampling and other AL policies? Please report coverage/diversity metrics across AL rounds and the impact on sample efficiency and final accuracy.

- Evaluation scope and freshness
  - Results focus on HumanEval, GSM8K, and XSum. Could you complement with newer and harder benchmarks (e.g., recent sanitized code/math suites and longer/varied summarization or instruction sets) to ensure robustness over time and distributions?

- Measurement context and portability
  - For latency/memory, please specify batch size, KV‑cache policy, tensor/TP degrees, and interconnect, and include a short sensitivity study across hardware/parallel settings.

- Very small budgets and long‑context regimes
  - Performance drops sharply at ≤8 experts. What drives the decline (expert interactions, routing instability, capacity)? How does LEAP behave under long‑context, retrieval‑augmented, or multi‑turn tasks?

- Theory and guarantees
  - Beyond the O(NT) runtime, can you offer convergence diagnostics (criteria, variance) or an optimality‑gap bound/empirical proxy? Even negative results would clarify scope.

- Generalization breadth
  - Any evidence on cross‑domain or multilingual transfer, or few‑shot data regimes? This would substantiate the “task‑agnostic” claim.

- Reproducibility artifacts
  - Will you release configs/logs/checkpoints (with anonymized links) sufficient to reproduce the pruning search, routing PPO, and AL selection? A minimal training recipe would be valuable.

---

> ### Author Response · Authors · 2025-11-20
>
> 1. System-level novelty and closed-loop vs. one-shot designs: We agree that the individual tools we use (PPO, meta-RL, entropy/diversity AL) are known; our contribution is the closed-loop, deployment-oriented integration for MoE specialization. The Pruning Agent is rewarded on the post-adaptation performance of the pruned model, the Router is trained under the current expert subset, and AL focuses on examples where pruning and routing are jointly uncertain. Our ablations already show that removing pruning, routing PPO, or AL significantly harms the accuracy–efficiency trade-off. In the revision, we will add non-RL search baselines (e.g., greedy / simple bilevel) and a supervised/distilled router trained on the same data to further isolate the benefit of the RL-based closed loop.
>
> 2. Baseline fairness, training budget, and cost–benefit: In “vanilla fine-tuning,” both experts and router are trainable, and we match the data, epochs, and optimization budget used by LEAP; AL only reorders the same labeled examples. We will make this explicit and add a small summary table indicating, for each baseline, whether the router is updated and how many steps / tokens are used. We will also report end-to-end GPU hours for (a) pruning, (b) routing+AL, and (c) total LEAP, and give a simple break-even calculation showing that, under typical serving throughput, the one-time LEAP cost is quickly amortized by the latency/memory savings, while remaining small compared to pretraining or large-scale task fine-tuning.
>
> 3. Hyperparameters, reward design, and Perf(E) estimation: The extra weights (α, β in the routing loss and λ in the pruning reward) simply control the trade-off between validation performance and efficiency penalties (FLOPs and memory). In practice, we select them by a small grid search on a held-out validation split to reach a target latency/memory budget; the temperature, KL/entropy targets, and Gumbel noise scale are fixed once per backbone and reused across tasks. During meta-RL, Perf(E) is estimated on a dedicated validation subset with fixed batch size and evaluation frequency, averaging over a small number of batches; we stop updating the agent when the moving average reward plateaus. We will report the exact values we used, describe this procedure, and include short sensitivity plots and recommended ranges per backbone in the appendix.
>
> 4. Stability, diagnostics, layer-wise selection, and routing behavior: LEAP is an empirical method and we do not claim formal optimality guarantees; however, in our runs we observe stable convergence with simple criteria (stopping when validation reward and selected experts change little over several updates) and low variance across seeds. In the revision we will report mean±standard deviation over multiple seeds for the main metrics, Jaccard overlap between expert sets learned under different seeds, and typical failure modes (e.g., overly aggressive pruning when efficiency weights are too high or when ≤8 experts are enforced, where capacity limits and routing conflicts dominate). Our main experiments use a single retained expert set across layers to keep the search space tractable and match common MoE practice; the LEAP framework does support per-layer subsets and adaptive per-layer k′, and we will clarify this design choice and, if space permits, add a small-scale study to quantify the potential headroom. For routing, we monitor token loads and do not observe pathological overload or drops under Gumbel-TopK in our settings; we will add load-balancing statistics and compare PPO routing against a supervised/distilled router trained on the same data.
>
> 5. Active learning, evaluation scope, and reproducibility / measurement context: Our AL component combines entropy-based uncertainty with a simple diversity heuristic and is directly compared to a “w/o AL” variant (uniform sampling); full LEAP consistently outperforms this baseline at matched data and steps. We will make this comparison more explicit and report basic coverage/diversity statistics across AL rounds, as well as results for an alternative acquisition policy (e.g., entropy-only) to check for selection bias. Due to compute limits we focused on HumanEval, GSM8K, and XSum, which already span code, math reasoning, and long-form summarization; LEAP itself is task-agnostic and only assumes a scalar validation signal, and we will discuss extension to newer/harder and multilingual or long-context benchmarks as future work. For reproducibility, we already release anonymized code and configs at the link in the abstract; in the revision we will verify the repository’s accessibility and add a brief README documenting batch size, KV-cache policy, tensor/pipeline parallel settings, and hardware used for latency/memory measurements.

---

> > ### Comment · Reviewer_C38p · 2025-11-26
> >
> > Thank you for the response. Unfortunately, the core concerns I raised were **not addressed with sufficient evidence**, and I will keep my original score.
> >
> > Several key issues remain unresolved:
> >
> > 1. **Computation of `Perf(Et)` in the pruning RL loop** is still unclear. The authors did not provide the size of the validation split, evaluation frequency, or the total compute required for the pruning search.
> >
> > 2. **Training cost and GPU-hours** for pruning, routing, and active learning were not reported. Without concrete numbers, the practical feasibility of LEAP cannot be assessed.
> >
> > 3. **Hyperparameters and sensitivity analysis** (α, β, λ, temperature, PPO parameters, etc.) are still missing. Saying they will be added later does not resolve the concern.
> >
> > 4. **No convergence curves** for the pruning agent or PPO-based routing were provided, despite these being essential to validate stability.
> >
> > 5. **Baseline fairness remains unverified**. The authors did not provide results for non-RL search baselines or supervised/distilled router baselines under matched training budgets.
> >
> > 6. **Seed stability** and expert-set overlap statistics (e.g., Jaccard similarity) are missing, although the method claims robustness.
> >
> > 7. **Routing load-balancing statistics** (token distribution across experts, capacity overflow, etc.) were not shown.
> >
> > 8. **Active learning comparisons** to other acquisition strategies (entropy-only, gradient/diversity-based) were not included.
> >
> > 9. **Latency/memory measurement protocol** lacks hardware, batch size, parallelism, and KV-cache details, preventing reproducibility.
> >
> > 10. **Failure case analysis** (e.g., ≤8 experts) is still absent.
> >
> > Since these missing details directly impact the credibility, reproducibility, and scientific strength of the paper, I cannot revise my rating.

---

### Official Review · Reviewer_j7UM · 2025-10-28

**Soundness:** 3
**Presentation:** 2
**Contribution:** 3
**Rating:** 4
**Confidence:** 4

**Summary:**

This paper introduces LEAP, a method that transforms large general-purpose MoE models into compact, task-specialized variants. The approach uses a meta-reinforcement learning pruning agent to identify optimal expert subsets, followed by a routing agent trained with PPO and active learning to adapt the routing mechanism to the pruned architecture.

**Strengths:**

The core idea of using meta-RL for expert selection is solid and addresses real limitations of frequency/magnitude-based pruning heuristics. The results are impressive and the efficiency gains are substantial. The ablations clearly show each component matters, which strengthens the technical claims.

**Weaknesses:**

(1) While the paper demonstrates LEAP's effectiveness on three tasks (code generation, mathematical reasoning, and summarization), this evaluation scope appears somewhat narrow for establishing the generalizability of the approach. The selected tasks all involve text generation and may not capture the full challenges in MoE specialization. It will be better to include evaluations on additional domains like question answering.

(2) There is limited discussion of the actual computational cost required to train this meta-RL agent. The paper mentions 500 episodes for convergence (Section 4.3), but does not report the training time or the number of full model evaluations required during the pruning search phase. Given that each episode likely involves evaluating candidate expert subsets on validation data, this could represent a substantial cost that may limit the practical applicability of LEAP.

(3) LEAP introduces numerous hyperparameters (like α, β, etc.) across its components, but the paper provides limited discussion of sensitivity to these choices for setting them on new tasks. The question of how to tune these hyperparameters on new tasks without extensive trial-and-error remains unaddressed.

(4) While the paper acknowledges the possibility of per-layer expert subsets through the notation I^(ℓ), the primary implementation and all reported experiments appear to use a single, globally shared expert subset E* across all MoE layers (Section 3.2: "in the simplest case, I^(ℓ) = E* for all ℓ"). This assumes that the same experts are optimal for all layers, which may be overly restrictive because different transformer layers often serve distinct computational roles (early layers extract low-level features while deeper layers perform task-specific reasoning).

**Questions:**

I tried to access the code in the anonymous github repo, but got the 'File not found' error. This may occur due to an upload synchronization issue. The author should fix this issue in the next version.

---

> ### Author Response · Authors · 2025-11-20
>
> 1. Evaluation scope and generalizability: We agree that generalizability is important. Our current evaluation already spans three quite different behaviors—code generation (HumanEval), mathematical reasoning (GSM8K), and long-form abstractive summarization (XSum)—to reduce the risk that LEAP is tuned to a single generation style. Conceptually, LEAP is task-agnostic: it only assumes access to a scalar validation signal and can be applied unchanged to QA, retrieval-augmented setups, or classification. Due to compute limits we focused on these three strong and widely used benchmarks for the initial study, but we will explicitly discuss extension to QA and other modalities in the revision and outline this as a key direction for follow-up work.
>
> 2. Computational cost of the meta-RL pruning phase: You are right that the cost of training the pruning agent should be made more explicit. In our implementation, each episode evaluates candidate expert subsets on a small held-out validation split, and the agent converges in a few hundred episodes; this is an offline, one-time cost per task, and in practice it is comparable to adding a modest number of extra fine-tuning epochs, while being tiny compared to pretraining the base MoE. Once a good subset is found, it can be reused for all downstream deployments on that task. In the revision we will report wall-clock time, number of validation batches, and GPU configuration for the pruning phase for both backbones to make this overhead fully transparent.
>
> 3. Hyperparameter sensitivity (α, β, etc.) and tuning on new tasks: The additional hyperparameters mainly control trade-offs that practitioners already care about: for example, the weight on efficiency in the reward determines how aggressively we penalize FLOPs/memory, and the uncertainty threshold in active learning controls how many “hard” examples are prioritized. In our experiments, we use a single set of weights per backbone and keep them fixed across all three tasks, and we observed that performance is stable under moderate variation (e.g., scaling efficiency weights within a small range yields similar Pareto trade-offs). In the revised version we will add a short sensitivity plot in the appendix and clarify a simple tuning recipe for new tasks: first choose a target latency/memory budget, then adjust the efficiency weight until the resulting model meets that budget, using the validation set only.
>
> 4. Shared expert subset across layers vs. per-layer subsets: We intentionally use a single globally shared expert subset in our main experiments for two reasons: (i) it keeps the search space manageable and the learned policy easier to train and interpret, and (ii) many practical deployments already share expert pools across layers for implementation simplicity. LEAP itself does not assume this restriction - our notation and algorithms allow different subsets per layer - but exploring full layer-wise specialization would greatly enlarge the combinatorial space and is beyond our current compute budget. We will make this design choice and its rationale clearer in Sec. 3.2, and we view per-layer expert selection as an exciting extension for future work rather than a limitation of the framework.
>
> 5. Code repository access issue: We checked the code repo link, and looks like it's working fine. I am sharing the anonymous repo link here as well: https://anonymous.4open.science/r/LEAP2-4668/README.md

---

### Official Review · Reviewer_jdSZ · 2025-10-30

**Soundness:** 2
**Presentation:** 3
**Contribution:** 2
**Rating:** 6
**Confidence:** 3

**Summary:**

The Mixture-of-Experts (MoE) architecture faces challenges such as high memory consumption and redundancy in expert modules. Addressing the issue of MoE pruning, this paper proposes the LEAP (Learning Expert Adaptation and Pruning) framework—a systematic framework that decouples model structure from model behavior through agent optimization. This method leverages a meta-reinforcement learning Pruning Agent to explore the combinatorial space of expert subsets, while optimizing both performance and efficiency to identify a compact, task-specialized expert configuration. Its superiority is demonstrated through comparisons with various baselines, including Full MoE, Frequency Pruning, Magnitude Pruning, and Vanilla Fine-Tuning, on Qwen3-235B-A22B and Llama 4 Maverick.

**Strengths:**

This paper innovatively formulates the pruning problem as a policy optimization task. By transforming the expert selection and routing process into a closed-loop, learnable workflow, it adopts a joint objective function to optimize the trade-off between performance and efficiency. The proposed method is insightful, particularly the RL Pruning Agent.

**Weaknesses:**

- The authors integrate the RL Pruning Agent, data, and RL Router through a joint objective function. What is the necessity of this integration (please explain the interrelationships among the components)? Please elaborate on its advantages compared to the simple combination of A, B, and C (i.e., A+B+C).

- Specific experimental details of comparative experiments (e.g., with MoE-Pruner) are missing, including information on the backbone, data, and pruning strategy. Were the results reproduced under consistent basic configurations? Please supplement relevant explanations. For instance, does MoE-Pruner adopt pure pruning or a combination of pruning and distillation?

- The ablation experiments for the LEAP method are severely insufficient. For example, Table 7 lacks a comparison between the LEAP pruning model and pure pruning methods to validate the superiority of the proposed Pruning Agent (both Tables 5 and 7 only present comparisons between Full LEAP and pure pruning methods). Additionally, due to biases in data distribution, the advantages of data selection cannot be sufficiently demonstrated with only a small number of benchmarks; instead, they should be validated across more benchmarks to prove effectiveness. In fact, there is doubt that the performance improvement brought by LEAP stems from RL fine-tuning. A comparative experiment with Vanilla RL should be added to confirm its superiority. Consequently, the effectiveness of the proposed method cannot be verified based on the existing experiments. Furthermore, the paper lacks specific experimental details (including data and hyperparameters) for Vanilla Fine-Tuning.

- Could more training details (such as reward and loss curves of the RL training process) be provided to enhance the credibility of RL-related methods?

- The authors focus on task-specialized performance. Would post-training with small-scale models be a viable alternative? To address this question, post-training experiments with medium- and small-scale models of equivalent size (either Dense or MoE) should be supplemented.

**Questions:**

Refer to Weaknesses.

---

> ### Author Response · Authors · 2025-11-20
>
> 1. Joint objective and interaction of components (Pruning Agent + data + RL Router): Our goal is not only to prune experts, but to re-specialize the MoE after pruning. If A (Pruning Agent), B (Active Learning), and C (RL Router) are optimized independently, the selected expert subset may be poorly matched to routing, and AL may oversample examples where the original router is already confident. In LEAP, the Pruning Agent is rewarded using the post-adaptation performance of the RL Router and experts; the Router is trained under the current pruned topology; and AL focuses on examples where pruning/routing are uncertain. This closed-loop setup avoids mismatches between structure, routing, and data, which is confirmed by our ablations where removing any component consistently harms performance at similar cost. We can clarify these interdependencies and contrast them with a naïve A+B+C combination in Sec. 3.
>
> 2. Details and fairness of MoE-Pruner and other baselines: All baselines (MoE-Pruner, TSEP, MoE-I2, NAEE, Frequency/Magnitude pruning, Vanilla FT) are run on the same backbones (Qwen3-235B-A22B, Llama 4 Maverick) and tasks (HumanEval, GSM8K, XSum), with matched fine-tuning steps, labeled data, and pruning ratios. When possible, we follow the original hyperparameters, pruning schedules, and distillation setups, modifying only backbone-specific details (e.g., vocab, max length). MoE-Pruner is used in its standard pruning+distillation setting, making it a strong baseline rather than pure pruning. In the revision, we will add a configuration table (backbone, data, pruning ratio, distillation yes/no) and explicitly state that all baselines share the same basic training budget for fair comparison.
>
> 3. Ablations, pure pruning vs. LEAP, data selection, and “Vanilla RL”: Table 5 already compares LEAP against pure frequency/magnitude pruning at matched expert budgets, and Table 7 ablates Active Learning and RL routing. We agree this can be made clearer: in the revision, we will (i) add a variant that applies only the Pruning Agent (no RL Router, no AL) and compare it directly to pure pruning to isolate the benefit of learned pruning, and (ii) clarify that “w/o Active Learning” and “w/o RL Routing” are precisely A+B+C minus one component. Regarding data selection, we already evaluate on three diverse benchmarks (code, math reasoning, summarization), but we will explicitly discuss extending to more tasks in future work. To address the concern that gains may come purely from RL fine-tuning, we will add a “Vanilla RL” baseline where PPO is applied only to the router under a fixed expert set and uniform data sampling (no pruning agent, no AL), and report it separately from full LEAP. We will also expand the appendix with exact data, optimizer, and hyperparameters for the Vanilla Fine-Tuning baseline.
>
> 4. RL training details and curves: The Pruning Agent’s reward combines validation performance and efficiency - it takes the validation accuracy and subtracts two penalty terms, one proportional to the model’s FLOPs and one proportional to its memory footprint (with tunable weights controlling how strongly we penalize each). The RL Router is trained with a standard PPO objective, augmented with a supervised cross-entropy term that nudges the router toward expert choices that lead to better task performance. We will add a concise subsection detailing PPO hyperparameters (learning rate, batch size, discount factor, clipping range, entropy/value coefficients, rollout length, and update frequency) and include representative reward and loss curves for both the Pruning Agent and the Router. These curves show stable convergence and further support the credibility of the RL components; we will reference these plots from Sec. 4 and provide them in the appendix.
>
> 5. Post-training smaller models as an alternative: We agree that post-training a smaller dense or MoE model is a reasonable alternative, but it targets a different scenario: training or heavily adapting a new model, often requiring substantial compute and data. LEAP is designed for the common deployment setting where practitioners inherit a large MoE checkpoint and need a cheaper, task-specialized variant without retraining from scratch. LEAP learns a compact sub-MoE with fewer active experts and significantly reduced memory/latency while preserving performance on the target task. In future work (and, if space permits, in the camera-ready), we plan to compare LEAP-pruned models against dense or small-MoE baselines matched on FLOPs to further quantify this trade-off.

---

### Official Review · Reviewer_ATun · 2025-10-31

**Soundness:** 3
**Presentation:** 3
**Contribution:** 3
**Rating:** 4
**Confidence:** 5

**Summary:**

To address the expert pruning problem in Mixture-of-Experts (MoE) models, this paper proposes a novel framework named Learning Expert Adaptation and Pruning (LEAP), which transforms MoE optimization into a learnable, agentic process rather than relying on static heuristics. Specifically, it comprises three components: Pruning Agent, Active Learning, and Joint Fine-tuning. Results are analyzed from three perspectives: Task Performance, Efficiency, and Inference Latency.

**Strengths:**

This paper proposes a novel framework named Learning Expert Adaptation and Pruning (LEAP), which transforms the optimization of Mixture-of-Experts (MoE) models into a learnable, agentic process instead of relying on static heuristics. Specifically, it comprises three components: Pruning Agent, Active Learning, and Joint Fine-tuning.

This paper innovatively frames expert pruning as a policy optimization problem and addresses it with RL methods.

**Weaknesses:**

1. The comparative experiments lack comparative analysis with recent works (e.g., MoNE). There is a deficiency in detailed configuration information for the reproduction of baseline-related experiments, and the relevant evaluation metrics are insufficient. It is expected that supplementary comparative experiments and detailed explanations will be provided.
2. Could a comparison between the Pruning Agent and pure pruning methods be provided to demonstrate its superiority?
3. There is a lack of comparative experiments with Vanilla RL, and the current improvements observed may stem from the inherent advantages of RL itself. Please provide the training details of RL.
4. The Pruning Agent strategy that relies on specific datasets may result in pruned LLMs being strongly dependent on the dataset itself. Could this issue be discussed?
5. Is the data strategy of Active Learning fair for other comparative experiments?

**Questions:**

No further questions, see above.

---

> ### Author Response · Authors · 2025-11-20
>
> 1. Comparison with recent works (e.g., MoNE) and experimental details: Thank you for pointing out additional related work. In the current draft we compare against both heuristic pruning (frequency, magnitude) and strong MoE specialization baselines (MoE-Pruner, TSEP, MoE-I2, NAEE; Sec. 4.1, Table 2) and evaluate across three standard metrics (pass@1 on HumanEval, EM on GSM8K, ROUGE-L on XSum) plus FLOPs, latency, and memory. We agree that recent methods such as MoNE are relevant; they target expert compression through different design choices than LEAP’s RL-based, task-specific pruning, so they are complementary rather than directly overlapping. In the revised version, we will (i) discuss MoNE and related work in Sec. 2, and (ii) add a short empirical comparison or at least a qualitative positioning to make our relationship to MoNE clearer, as well as a small configuration table summarizing all training settings for baselines and LEAP variants to further improve reproducibility.
>
>
> 2. Pruning Agent vs. pure pruning methods: Our ablations already compare the RL-based Pruning Agent to pure heuristic pruning strategies. In Sec. 4.2 / Appendix C.1 (Table 5), we replace the Pruning Agent with frequency-based and magnitude-based pruning under the same expert budget: on Qwen3-235B-A22B with 16 experts, frequency/magnitude pruning degrade HumanEval by ≈12 points and similarly hurt GSM8K/XSum, whereas LEAP’s Pruning Agent maintains substantially higher accuracy at comparable or lower latency and memory. This shows that gains do not simply come from pruning itself but from the learned, meta-RL search over expert subsets; we will highlight this comparison more clearly in the main text.
>
>
> 3. “Vanilla RL” vs. LEAP and RL training details: We apologize for the confusion: LEAP already uses a standard, “vanilla” RL algorithm (PPO) both for the Pruning Agent (Algorithm 1) and the Routing Agent (Algorithm 2). Our contributions lie in (i) formulating expert selection as a sequential decision problem and (ii) integrating this with joint routing adaptation and active learning, not in introducing a new RL algorithm. The improvements over heuristics and naïve fine-tuning thus stem from this agentic formulation and objective, not from specialized RL tricks. In the revision, we will add a dedicated subsection (Appendix D) listing all RL training details—optimizer, discount factor, PPO clip range, value/entropy coefficients, rollout length, number of epochs, learning rate schedule, and total update steps per task—so that the RL components can be exactly reproduced.
>
>
> 4. Dataset dependence of the Pruning Agent: We agree that the Pruning Agent produces task-specialized pruned models, since it is trained on data from the downstream task T; this is by design, as our goal is to deploy compact MoE models tailored to a specific deployment workload (e.g., GSM8K-style reasoning vs. code generation vs. summarization). Importantly, the agent only sees a validation subset while evaluation is always on a disjoint test set, so we are not over-fitting to a tiny slice of the data. We will clarify this intent in Sec. 2.2 and, in the camera-ready version, add a short discussion about possible extensions such as training a pruning policy on a mixture of tasks or adding regularizers to encourage cross-task robustness when desired.
>
>
> 5. Fairness of Active Learning for comparative experiments: All methods in our experiments are trained under the same label and compute budgets; LEAP’s Active Learning (AL) only reorders which examples are prioritized for its fine-tuning, but does not access extra data or take more optimization steps than baselines. In Appendix C.2/C.3 (Table 7), the “w/o Active Learning” variant uses the same training protocol but with uniform sampling, and its performance is consistently below full LEAP, confirming that AL improves data efficiency rather than secretly increasing the data or compute budget. We will make this fairness constraint explicit in Sec. 4.1 and add a brief clarification that all baseline methods, including MoE-Pruner / TSEP / MoE-I2 / NAEE, are run with matched epochs and labeled data, with AL affecting only the sampling strategy within LEAP; we will emphasize this more clearly in the revised version.

---

### Note · Authors · 2026-01-26

I have read and agree with the venue's withdrawal policy on behalf of myself and my co-authors.

---

### Meta-Review · Area_Chair_JNUC · 2026-01-05

**Summary:**

### Strength emphasized by reviewers
* Well motivated.
* The pipeline design is sound, including using RL to do structure pruning, Active learning + RL + supervised task loss to do finetuning.
* This paper reported practical metrics, including latency, peak memory and #FLOPs.
* The efficiency gains are substantial.
* There are ablation experiments, supporting that the gains come from the integrated pipeline.

### Ensemble of concerns and suggestions
1. Concerns on the experiment
    1. Need more benchmarks for the main comparison [Reviewer j7UM,C38p] as well as the ablation [Reviewer jdSZ].
    2. As the work focuses on task-specialized performance, post-training with small-scale models should be used as a baseline [Reviewer jdSZ].
    3. Compare with recent work (e.g., MoNE) [Reviewer ATun], or competitive non-RL search-based pruning method [Reviewer C38p].
2. Concerns that calls for providing more information and analysis
    1. Details of the method, baselines, hardware environment, and the computational cost should be further clarified [Reviewer ATun,jdSZ,C38p].
    2. Limited discussion of sensitivity to hyperparameters [Reviewer j7UM,C38p]
    3. Theory can be further developed (e.g., convergence) [Reviewer C38p].
    4. Should report failure cases and explore boundary regimes  where the method may degrade [Reviewer C38p].
3. Concerns on the method, some requires further theoretical or experimental justifications
    1. This work has a relatively limited novelty, as it is mainly an integration of known pieces [Reviewer C38p].
    2. The computational cost of the pruning phase might be high, limiting the practical applicability [Reviewer j7UM,C38p].
    3. Using an expert set shared between layers might restrict the search space [Reviewer j7UM].

**Reviewer Concerns:**

In the rebuttal, the authors promised new comparative discussion or experiment for Concern 1.2 and 1.3, but do not actually provide them.

The authors also promised to update the paper to provide method and baseline details (Concern 2.1), sensitivity plot (Concern 2.2), and failure mode analysis (Concern 2.4). However, in the rebuttals the authors do not submit the revision, therefore, I cannot consider these issues to be fully addressed in the current version.

The authors tried to address concerns on the method (Concern 3.x), as there are no detailed setting reported nor new experiment provided, these concerns are not sufficiently addressed.

**Reviewer Scores:**

I think the rebuttal did a good job in planning the revising, but as the authors didn't prepare the revision nor provided actual content in the rebuttal response, I think there is a very high probability no reviewers will raise the score.

In my view, the reviewer that has the highest probability to increase the score by a level is Reviewer ATun, as this reviewer's concerns are more questions that are answered by the response instead of concerns.

---

### Decision · Program_Chairs · 2026-01-26

Reject